# UNDERSTANDING DATA POISONING ATTACKS FOR RAG: INSIGHTS AND ALGORITHMS

## ABSTRACT

Large Language Models (LLMs) have achieved success across various domains but also exhibit problematic issues, such as hallucinations. Retrieval-Augmented Generation (RAG) effectively alleviates these problems by incorporating external information to improve the factual accuracy of LLM-generated content. However, recent studies reveal that RAG systems are vulnerable to adversarial poisoning attacks, where attackers manipulate retrieval systems by poisoning the data corpus used for retrieval. These attacks raise serious safety concerns, as they can easily bypass existing defenses. In this work, we address these safety issues by first providing insights into the factors contributing to successful attacks. In particular, we show that more effective poisoning attacks tend to occur along directions where the clean data distribution exhibits small variances. Based on these insights, we propose two strategies. First, we introduce a new defense, named DRS (Directional Relative Shifts), which examines shifts along those directions where effective attacks are likely to occur. Second, we develop a new attack algorithm to generate more stealthy poisoning data (i.e., less detectable) by regularizing the poisoning data's DRS. We conducted extensive experiments across multiple application scenarios, including RAG Agent and dense passage retrieval for Q&A, to demonstrate the effectiveness of our proposed methods.

## 1 INTRODUCTION

Large Language Models (LLMs) have demonstrated impressive performance across various benchmark tasks in many domains (Achiam et al., 2023; Thirunavukarasu et al., 2023). However, LLMs can also exhibit several problematic behaviors, such as hallucinations (Ji et al., 2023) and bias (Bender et al., 2021), which could possibly lead to dire consequences when applied in safety-critical areas like healthcare (Tian et al., 2024). To address these issues, Retrieval-Augmented Generation (RAG) (Khandelwal et al., 2019; Lewis et al., 2020; Borgeaud et al., 2022; Ram et al., 2023) has been introduced as a promising approach that integrates external knowledge into LLM outputs, offering a potential solution.

Typical RAG operates through two procedures: retrieval and generation. When a query is received, it first retrieves relevant information from an external data source, such as Wikipedia, and then combines this retrieved nonparametric knowledge (i.e., external knowledge) with the LLM's parametric knowledge (i.e., internal knowledge) to generate the final output. Extensive empirical evidence suggests that LLMs leveraging RAG can effectively reduce hallucination (Bender et al., 2021; Kirchenbauer & Barns, 2024; Li et al., 2024a) and improve prediction accuracy in knowledge-intensive domains such as finance and medicine (Borgeaud et al., 2022; Xiong et al., 2024). For example, research by (Kirchenbauer & Barns, 2024) incorporated a RAG system into the Mistral-family LLMs and observed significant improvements in factual accuracy.

Despite the aforementioned benefits of RAG, a line of very recent work has demonstrated that RAG systems are vulnerable to adversarial poisoning attacks across multiple application scenarios (Zou et al., 2024; RoyChowdhury et al., 2024; Chen et al., 2024b;a; Tan et al., 2024; Shafran et al., 2024; Xue et al., 2024; Cheng et al., 2024). In these attacks, malicious attackers exploit the openly accessible nature of the database corpus used for retrieval in RAG—such as Wikipedia (Zou et al., 2024; Deng et al., 2024). By injecting attacker-specified data into the corpus, attackers can manipulate the retriever to return the poisoned data as the most relevant documents in response to attacker-specified queries, thereby increasing the chance that LLMs will generate adversarial outputs when relying on the poisoned data.

These RAG attacks can be characterized as *data poisoning attacks against retrieval systems*. As summarized in Table 1, they all involve poisoning the data corpus used for retrieval, even though the attacks may differ in their level of access to the retrievers and/or LLMs. The ultimate goal of all these attacks remains the same: to have the injected poisoned data retrieved in response to attacker-specified queries, thereby influencing the subsequent LLM generation. On the one hand, categorizing these RAG attacks as data poisoning is promising, given the extensive body of research on defending against such attacks. On the other hand, data poisoning attacks, like those in the computer vision literature, remain difficult to defend against (Liang et al., 2022). This naturally raises a question: are data poisoning attacks against retrieval systems just as difficult to defend against as data poisoning attacks in computer vision?

Several recent studies have demonstrated that existing state-of-the-art defenses can be ineffective against these attacks. For instance, perplexity-based filters, which examine the perplexity of documents and flag those with abnormally high or low perplexity values, have proven ineffective for detecting poisoned documents (Chen et al., 2024a; Zou et al., 2024). Given the increasing use of RAG systems in safety-critical domains like healthcare, it is crucial to identify why current defenses are failing and to design new approaches to ensure their secure implementation.

In this work, we investigate the safety vulnerabilities in RAG systems, focusing on targeted data poisoning attacks aimed at retrieval systems. We begin by analyzing why these attacks are effective (in terms of attack success rates) and difficult to defend against. Building on this analysis, we introduce two designs: (1) a novel algorithm to mitigate these attacks, and (2) a method for generating adversarial poisoning data that is harder to detect. Our main contributions are summarized as follows.

- **Insights into understanding the effectiveness of targeted data poisoning attacks against retrieval systems.** There are two types of data poisoning attacks for RAG: (i) targeted attacks and (ii) untargeted attacks. Targeted attacks refer to attacks aimed specifically at a set of attacker-specified data (e.g., pre-selected questions (Zou et al., 2024)), while untargeted attacks aim to affect all data. We focus on targeted attacks, which make up most data poisoning attacks (see Table 1), as previous research has shown that untargeted attacks can be effectively mitigated using existing methods (Zhong et al., 2023). First, we demonstrate that these targeted attacks can be characterized using a common objective function(s). This formulation allows us to provide both quantitative and qualitative insights into the question: What are the most effective methods for conducting these attacks? In particular, we show that more effective attacks should occur along the directions where the clean data distribution (to be specified) diminishes most rapidly.

- **Derivations from developed insights (I): a new defense method against retrieval poisoning attacks.** Based on developed insights, we propose a new metric, dubbed DRS (Directional Relative Shifts), along with a corresponding filter-based defense utilizing the proposed DRS. Specifically, the DRS (to be defined) measures the relative shifts of future test documents that occur along the directions of clean documents with low eigenvalues. If the DRS score of a future test document is sufficiently abnormal compared to those of clean documents, we will flag this particular document as a poisoned one.

- **Derivations from developed insights (II): new attack algorithms for designing more stealthy poisoning data (in terms of detection).** We found that our proposed DRS defense can effectively distinguish the poisoned data generated by most existing attacks from clean data, motivating us to develop new algorithms capable of bypassing this defense. We introduce a regularization-based approach aimed at producing more stealthy poisoned data. In detail, we incorporate a regularization term into the original objective functions for optimizing to generate poisoned data, which penalizes large DRS values. By utilizing this regularization technique, the poisoned data created under this framework is more likely to bypass our DRS defense.

- **Extensive empirical study across different setups.** We test both our proposed defense and attack algorithms in various setups. **Defense**: The proposed DRS defense is evaluated across different RAG application scenarios: (1) RAG LLM-Agent (Chen et al., 2024a), (2) dense retrieval systems for general QA (Long et al., 2024), and (3) medical RAG applications (Zou et al., 2024). Our method significantly outperforms existing state-of-the-art approaches across many cases. **Attack**: We apply our new attack algorithms to generate more stealthy

Table 1: Summary of existing RAG attacks (involving the retrieval system) and adversarial attacks against dense retrieval systems. The second column indicates whether a certain attack was originally designed to attack the RAG system as a whole (denoted by RAG) or solely the dense retrieval system (denoted by DR). The third column indicates whether an attack is targeted or not. Here targeted attacks refer to attacks that are aimed at a particular subset of data, rather than indiscriminately affecting the entire dataset. In the fourth row, retriever access indicates whether the attack requires white-box (W) or black-box (B) access. The last column shows whether the proposed method requires access to the LLM. We can observe that almost all attacks are *targeted* attacks.

| | Attack Method | Poisoning Data Corpus | Targeted Attack | Retriever Access | LLM Access |
|---|---|---|---|---|---|
| **Agent Poison (Ap)** (Chen et al., 2024a) | RAG | ✓ | ✓ | W | ✓ |
| **Glue pizza** (Tan et al., 2024) | RAG | ✓ | ✓ | W | ✓ |
| **PoisonedRAG** (Zou et al., 2024) | RAG | ✓ | ✓ | W & B | ✗ & ✓ |
| **ConPilot** (RoyChowdhury et al., 2024) | RAG | ✓ | ✓ | W | ✓ |
| **Jamming** (Shafran et al., 2024) | RAG | ✓ | ✓ | B | ✗ |
| **BBox Opinion** (Chen et al., 2024b) | RAG | ✓ | ✓ | W | ✗ |
| **BadRAG** (Xue et al., 2024) | RAG | ✓ | ✓ | W | ✗ |
| **TrojanRAG** (Cheng et al., 2024) | RAG | ✓ | ✓ | W | ✗ |
| **CorpusPoi** (Zhong et al., 2023) | DR | ✓ | ✗ | W | NA |
| **Backdoor DPR** (Long et al., 2024) | DR | ✓ | ✓ | W | NA |
| **Contra DPR** (Liu et al., 2023b) | DR | ✓ | ✓ | B | NA |

red-teaming data in the RAG LLM-Agent scenario. The generated data maintain a similar level of attack success rate but can occasionally bypass our previously developed DRS defense, demonstrating the effectiveness of this newly developed attack method.

## 1.1 RELATED WORK

**Information retrieval** In recent decades, several key trends have emerged in the Information Retrieval (IR) literature. Classical sparse retrievers, such as BM25 (Robertson et al., 2009), rely on lexical matching and perform well when there is simple lexical overlap. However, in many domains, their performance lags behind that of dense retrievers (Zhao et al., 2024). Dense retrievers (Karpukhin et al., 2020; Izacard et al., 2021; Cohan et al., 2020), which leverage deep neural networks to match queries and documents based on semantic meaning, have demonstrated superior performance across a wide range of tasks (Zhao et al., 2024). One limitation of dense retrievers is that their components are often trained in isolation, which can negatively impact performance (Li et al., 2024b). To address this, generative retrievers have emerged, leveraging LLMs to generate relevant content in response to queries rather than retrieving documents (Bevilacqua et al., 2022).

**Adversarial attacks against dense retrieval systems** Dense retrieval (DR) systems have recently been shown to be vulnerable to a series of adversarial attacks (Liu et al., 2023b; Zhong et al., 2023; Long et al., 2024), which share similarities with black-hat search engine optimization techniques that have targeted traditional search engines (Gyongyi & Garcia-Molina, 2005). These adversarial attacks against DR systems share the same goal: manipulating the retriever to return attacker-crafted content, though their specific approaches differ. For example, (Liu et al., 2023b) assumes black-box access to the underlying retriever, using a surrogate model built to mimic the original system and craft poisoned data. In contrast, (Long et al., 2024) explores backdoor attacks under the assumption of white-box access, where attackers inject pre-specified query-response pairs into the training data, causing the retriever to return manipulated responses when presented with corresponding poisoned queries during inference.

**Adversarial attacks against RAG** The majority of existing RAG attacks focus on compromising retrieval systems with the goal of tricking them into retrieving adversarial documents (Zou et al., 2024; RoyChowdhury et al., 2024; Chen et al., 2024b;a; Tan et al., 2024; Shafran et al., 2024; Xue et al., 2024; Cheng et al., 2024). These attacks require varying levels of access to the retrievers and/or the LLMs, such as white-box (Tan et al., 2024) or black-box (Zou et al., 2024). However, all of these attacks need access to inject poisoned data into the underlying data corpus used by the RAG system, as summarized in Table 1. Additionally, almost all of them are targeted attacks, aimed at a particular subset of data, rather than indiscriminately affecting the entire dataset. In this sense, RAG attacks can essentially be regarded as *targeted data poisoning attacks* against the retrievers.

## 2 PRELIMINARY

**Notations.** We assume that dense retrievers are employed for retrieval in RAG following the convention. We denote $f$ as the embedding function (i.e., retriever) that takes text input (e.g., documents) and outputs its corresponding numerical representation, i.e., a $d$-dimensional real-valued vector. We denote the *clean* data corpus used for retrieval as $\mathcal{D}^{\text{clean}}$. We use $\ell_2$ distance as the similarity measurement. We use the notation $\mathcal{R}_k(q, \mathcal{D}, f)$ to represent the top-$k$ retrieved documents from a data corpus $\mathcal{D}$ corresponding to a query $q$ with the embedding function $f$. We employ the notation $\text{LLM}(q, \mathcal{R}_k(q, \mathcal{D}; f))$ to denote the outputs of the LLM based on query $q$ and the associated retrieved documents $\mathcal{R}_k(q, \mathcal{D}; f)$.

### 2.1 THREAT MODEL

**Attacker's Capability** Recall that there are two components in RAG: the retrieval system and the LLM. Regarding the retrieval system, we assume the attacker has white-box access, which naturally covers the black-box case (since black-box access is a more restricted form of white-box access). We assume the attacker can only inject poisoned data (e.g., by creating a new Wikipedia page), denoted as $\mathcal{D}^{\text{poi}}$, into the clean data corpus without modifying the original clean data corpus, $\mathcal{D}^{\text{clean}}$. This assumption is consistent with all existing RAG attack papers, as summarized in Table 1.

Meanwhile, attackers are assumed to have access to a set of target queries of interest. In terms of the LLM, we assume the attacker has only black-box access, i.e., they can obtain outputs but not modify the model itself, which is the realistic scenario for many proprietary models such as GPTs.

**Attacker's Goal:** Overall, the attacker aims to achieve the following two goals. **(a)** The RAG system should generate a prescribed adversarial output (e.g., a sudden stop for autonomous driving agents) in response to adversary queries. These queries could be deliberately crafted by the attacker, such as pre-selected questions (e.g., Who is the CEO of OpenAI? (Zou et al., 2024)) or sentences containing attacker-specified grammatical errors (Long et al., 2024). Formally, the attacker aims to maximize the adversary's performance objective:

$$\mathbb{E}_{\mathcal{D}^{\text{clean}}}\mathbb{E}_{\tilde{q}\sim Q_{\text{adv}}}\mathbf{1}\{\text{LLM}(\tilde{q}, \mathcal{R}_k(\tilde{q}, \mathcal{D}^{\text{poi}}\bigcup\mathcal{D}^{\text{clean}}; f) = \mathcal{S}_A\},$$

where $Q_{\text{adv}}$ is the distribution of adversarial queries $\tilde{q}$, $\mathcal{D}^A \triangleq \mathcal{D}^{\text{poi}}\bigcup\mathcal{D}^{\text{clean}}$ is the joint (poisoned) data corpus, $\mathcal{S}_A$ is the target malicious answer, and $\mathbf{1}\{B\}$ is the indicator function taking the value 1 if the event $B$ occurs and 0 otherwise.

**(b)** Ensure the outputs for clean queries remain unaffected. Formally, the attacker aims to maximize the normal performance

$$\mathbb{E}_{\mathcal{D}^{\text{clean}}}\mathbb{E}_{q\sim Q_{\text{normal}}}\mathbf{1}\{\text{LLM}(q, \mathcal{R}_k(q, \mathcal{D}^{\text{poi}}\bigcup\mathcal{D}^{\text{clean}}; f)) = \mathcal{S}_N\},$$

Here $Q_{\text{normal}}$ is the distribution of normal queries $q$, and $\mathcal{S}_N$ denotes the benign answer corresponding to a query $q$.

**Remark 1 (On preserving the normal utility).** We note that the second goal, namely preserving normal utility, is important. This differs from traditional untargeted attacks, such as those in (Zhong et al., 2023), which aim to degrade overall system performance. In fact, untargeted poisoning attacks, like those discussed in (Zhong et al., 2023), can be detected effectively using methods such as perplexity-based or $\ell_2$-norm-based defenses. Consequently, more recent research has shifted focus toward targeted attacks.

## 3 RAG ATTACKS UNVEILED: TARGETED DATA POISONING ATTACKS AGAINST RETRIEVAL SYSTEMS

In this section, we first demonstrate that achieving the two goals mentioned in the above section need to conduct targeted data poisoning attacks against the retrieval systems. Then we provide insights towards what leads to effective poisoning attacks against the retrieval systems.

We first observe that, given an adversarial query $\tilde{q}$, the LLMs will never output the attacker-prescribed outcome $\mathcal{S}_A$ if $\mathcal{R}_k(\tilde{q}, \mathcal{D}^{\text{poi}}\bigcup\mathcal{D}^{\text{clean}}; f)\bigcap\mathcal{D}^{\text{poi}}$ is empty. In other words, if the retrieved context $\mathcal{R}_k(\tilde{q}, \mathcal{D}^{\text{poi}}\bigcup\mathcal{D}^{\text{clean}}; f)$ does not contain any attacker-injected documents, the attacker-prescribed adversarial outcome will not occur. As a result, attackers are incentivized to ensure that all the

retrieved documents come from the poisoned data corpus $\mathcal{D}^{\text{poi}}$ they created and injected when querying the system with adversarial queries. This increases the chance that the system will generate the adversarial outputs they aim for. Formally, the attackers' first goal is:

$$\max \mathbb{E}_{\mathcal{D}^{\text{clean}}} \mathbb{E}_{\tilde{q} \sim Q_{\text{adv}}} \mathbf{1}\{[\mathcal{R}_k(\tilde{q}, \mathcal{D}^{\text{poi}} \bigcup \mathcal{D}^{\text{clean}}; f) \bigcap \mathcal{D}^{\text{clean}}] = \phi\}. \tag{1}$$

In a similar vein, for normal/clean queries to result in normal/benign answers from the LLMs, the retrieved content should exclude any poisoned data. Precisely, the attacker's second goal is to ensure that:

$$\max \mathbb{E}_{\mathcal{D}^{\text{clean}}} \mathbb{E}_{q \sim Q_{\text{normal}}} \mathbf{1}\{[\mathcal{R}_k(q, \mathcal{D}^{\text{poi}} \bigcup \mathcal{D}^{\text{clean}}; f) \bigcap \mathcal{D}^{\text{poi}}] = \phi\}. \tag{2}$$

### 3.1 THEORETICAL INSIGHTS TOWARDS UNDERSTANDING THE EFFECTIVENESS OF POISONING ATTACKS

In this section, we provide theoretical insights into understanding the effectiveness of attacks that satisfy the attackers' goals as specified in Eq. (1) and Eq. (2) respectively.

By taking a closer look at the attackers' dual-goal, there are a total of four components to consider: (i) the normal query set, (ii) the clean data corpus, (iii) the adversary query set, and (iv) the adversarial documents. It can be challenging to simultaneously consider the interactions between these components and their joint effect on the attackers' goals. As a result, we will first make the following assumptions regarding the adversary query distribution $Q_{\text{adv}}$ and the poisoned dataset $\mathcal{D}^{\text{poi}}$ to simplify and facilitate the analysis.

**Assumption 1.** (Closeness between adversary queries and adversary documents) For any small positive integer $k$ and $\tilde{q} \sim Q_{\text{adv}}$, there exists a subset $T \subseteq \mathcal{D}^{\text{poi}}$ with $|T| = k$, such that

$$\sup_{t \in T} \|\tilde{q} - t\| < \infty \quad a.s.$$

**Remark 2 (Intuitive understanding of Assumption 1).** Assumption 1 intuitively states that the poisoned documents should stay close to the adversarial query set. This assumption is realistic and easy to satisfy. For instance, the work (Zou et al., 2024) create poisoned documents $\mathcal{D}^{\text{poi}}$ directly appending poisoned text to the adversarial queries. Consequently, when querying with adversarial queries, these poisoned documents are likely to be retrieved, often with a top 1 ranking.

Assumption 1 essentially enables us to consider only the adversarial query $\tilde{q} \sim Q_{\text{adv}}$ without worrying about the adversarial documents, thus simplifying the overall analysis. Additionally, we assume that the distribution of the clean corpus $\mathcal{D}^{\text{clean}}$ has well-behave tailed, e.g., sub-Gaussian family of distributions. With these assumptions, we are now ready to formally state the first result.

**Theorem 1 (On the effectiveness of attacks).** Under the above assumptions, attackers' goals as specified in Eq. (1) and Eq. (2) can be met by using an adversary set $\tilde{q} \sim Q_{\text{adv}}$ that is sufficiently different from the distribution of the clean database $\mathcal{D}^{\text{clean}}$.

The above result can be intuitively interpreted as follows: The attacker aims to ensure that, when using adversarial queries, retrieved documents come mostly from the poisoned database. To achieve this, the attacker can create a set of adversarial queries that are significantly different from the clean ones, making the nearest neighbor documents entirely poisoned (as per Assumption 1). For illustration, consider a clean database consisting of texts about food, with the normal query set also focusing on food-related topics. The attacker could achieve their goal by using queries related to mathematics, which are irrelevant to the clean documents.

A potential *caveat* of the above argument is that if the adversarial queries are obviously different from the clean ones, they might be easily detected by simple human inspection. Therefore, to ensure that attacks remain effective under potential defenses, the attacker is more interested in the following question: ***Given the maximum deviation (e.g., $\ell_2$ distance) between the normal distribution $Q_{normal}$ and adversary distribution $Q_{adv}$, what is the most effective direction(s) for moving $Q_{normal}$ to $Q_{adv}$?***

**Corollary 1 (The most effective directions for attacks).** Under mild assumptions, the most effective directions for attacks, namely the directions maximizing Eq. (1) and Eq. (2), are the ones with the fastest decaying rates of the density of $\mathcal{D}^{\text{clean}}$.

**Remark 3 (Intuitions behind Corollary 1).** We provide a detailed interpretation of the above results. First, directions in $\mathcal{D}^{\text{clean}}$ whose density decays rapidly can typically corresponds to those directions with low variance across a broad range of distributions with well-behaved tails. Low variance in a given direction means that the majority of the probability mass is concentrated around the mean. Consequently, even a small deviation from the mean significantly reduces the probability mass in that area.

This behavior aligns well with the attacker's objectives. If a direction has low variance, perturbing a clean query along that direction will greatly decrease the likelihood of clean documents being near the perturbed query. As a result, adversarial queries will more likely retrieve poisoned documents, as the nearest neighbors around the query will predominantly be adversarial (according to Assumption 1). This indicates that the attacker's goals are effectively achieved.

In Figure 1, we provide empirical evidence to corroborate our theory. In particular, we follow the exact setup in (Chen et al., 2024a) to generate poisoned data from three different attacks (Ap, BadChain, and AutoDan). We can observe that the attack success rates of Ap are higher than BadChain and AutoDan. In other words, Ap attack is more powerful than others. We define the relative distances (i.e., relative mean along a direction divided by standard deviation along this direction) between adversarial documents and clean documents along the directions of clean documents with the top-100 smallest variance. We observed that more effective attacks, e.g., Ap, tend to have larger relative distance along these directions, which empirically verified our theory.

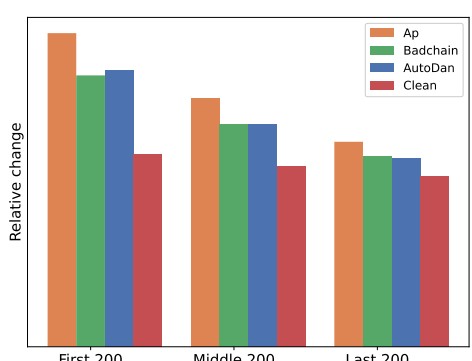

Figure 1: Empirical evidence towards verifying the developed theory. We plot the dimensional relative distances for three attacks—Ap, BadChain, and AutoDan (with Ap having a higher attack success rate than BadChain and AutoDan)—as well as clean documents, grouped by the scales of the standard deviation of each dimension. For example, the first 200 correspond to the 200 dimensions of clean document embeddings with the smallest standard deviations. We observe that more effective attacks (in terms of attack success rates) tend to have larger relative shifts along directions with small variances, corroborating our theory.

## 4 FROM DEVELOPED INSIGHTS: NEW DEFENSE AND ATTACK ALGORITHMS

In this section, building on our above developed insights, we propose two strategies: (1) a new detection-based defense for filtering out poisoned documents, and (2) a new attack algorithm designed to generate more stealthy poisoning data, namely, data that is less detectable.

### 4.1 NEW DEFENSE: DIRECTIONAL RELATIVE SHIFTS

We first outline the defense goal. The defender's goal is to protect a set of pre-selected queries of interest denoted as $\mathcal{Q}$, such as specific medical questions, from data poisoning attacks. Here, our defense aims preventing poisoned documents targeting these queries to get injected into the clean data corpus. We note the focus is on protecting a subset of targeted queries rather than all possible queries. As the number of queries increases significantly, the retrieved documents are likely to cover the entire text space. In those cases, the distinction between adversarial and benign queries becomes essentially indistinguishable.

Next, we describe the overall defense pipeline. We assume the defender has access to both the retriever and the clean data corpus. When a new test document is proposed for injection into the clean corpus, the defender calculates its DRS score (to be defined later in Eq. 3) and compares it with the scores of known clean documents. If the DRS score is abnormally large in comparison, the document will be flagged as potentially adversarial (pseudo code see Algorithm 2).

The above approach is motivated by our developed insights that identify a key feature of effective attacks. In particular, more effective attacks tend to cause larger shifts along directions where the variance is low. Therefore, we introduce the DRS (Directional Relative Shifts) metric to capture these shifts by measuring the distance between a test document and clean documents. If a test document

---

**Algorithm 1** Compute Directional Relevance Score (DRS)

---

**Input:** Standardized data matrix $\mathbf{X} \in \mathbb{R}^{n \times d}$, number of directions $M \leq d$

---

1: Perform eigendecomposition of the covariance matrix $\mathbf{S}$ of $\mathbf{X}$:

$$\mathbf{S} = \mathbf{V}\mathbf{\Lambda}\mathbf{V}^\top,$$

where $\mathbf{V} = \{\mathbf{v}_1, \ldots, \mathbf{v}_d\}$ contains eigenvectors and $\mathbf{\Lambda} = \mathrm{Diag}(\lambda_1, \ldots, \lambda_d)$ is the diagonal matrix of eigenvalues
2: Sort eigenvalues (and corresponding eigenvectors) in ascending order and denote the reordered index set as $\{\sigma(1), \sigma(2), \ldots, \sigma(d)\}$
3: Compute the DRS score for any $\mathbf{z}$ as:

$$\mathrm{DRS}(\mathbf{z}; \mathbf{X}) = \sum_{i=1}^{M} \frac{|\mathbf{z}^\top \mathbf{v}_{\sigma(i)}|}{\sqrt{\lambda_{\sigma(i)}}}, \tag{3}$$

where $\mathbf{v}_{\sigma(i)}$ is the $i$-th eigenvector corresponding to the $i$-th sorted eigenvalue $\lambda_{\sigma(i)}$

---

**Algorithm 2** Detection with DRS

---

**Input:** $\mathcal{Q}$: Set of targeted queries to be protected , $K$: number of documents to be retrieved, decision quantile $q \in (0, 1)$, a future test document $\mathbf{z}$

---

1: Retrieve top-$K$ *clean* documents for each query $q \in \mathcal{Q}$
2: Obtain embeddings of these retrieved documents $\mathbf{X}_{\text{clean}}$
3: Compute the DRS scores (by Algo. 1) for each $\mathbf{x} \in \mathbf{X}_{\text{clean}}$ denoted as $\{\mathrm{DRS}(\mathbf{x_i}; \mathbf{X}_{\text{clean}})\}_{i=1}^{|\mathbf{X}_{\text{clean}}|}$
4: Set the $\tau$ (decision threshold) to be $q$th quantile of $\{\mathrm{DRS}(\mathbf{x_i}; \mathbf{X}_{\text{clean}})\}_{i=1}^{|X_{\text{clean}}|}$
5: Reject a future document $\mathbf{z}$ if $\mathrm{DRS}(\mathbf{z}; \mathbf{X}_{\text{clean}}) > \tau$

---

**Output:** Decision on whether a future test document $\mathbf{z}$ is clean or adversarial.

---

is indeed adversarial, we expect it to have an excessively large DRS. Section 5 provides extensive empirical evidence on this.

We provide the detailed pseudo-code for calculating the DRS score in Algorithm 1 and the overall workflow for detection using the proposed DRS score in Algorithm 2, respectively. In detail, given $\mathcal{Q}$: a set of targeted queries to be protected, first, we (the defender) retrieve their associated *clean* top-$K$ documents and obtain their embeddings, denoted as $\mathbf{X}_{\text{clean}}$. We then compute the DRS scores for each $\mathbf{x} \in \mathbf{X}_{\text{clean}}$ as outlined in Algorithm 1. Next, we select the $q$th quantile, e.g., the 99th quantile, of the clean DRS scores $\{\mathrm{DRS}(\mathbf{x_i}; \mathbf{X}_{\text{clean}})\}_{i=1}^{|\mathbf{X}_{\text{clean}}|}$ to serve as a threshold for filtering out future poisoned documents. Given a future test document with embedding $\mathbf{z}$, we calculate its score and flag it as a poisoned sample if $\mathrm{DRS}(\mathbf{z}; \mathbf{X}_{\text{clean}}) > \tau$, where $\tau$ is the previously selected $q$th quantile.

### 4.2 NEW ATTACK ALGORITHMS FOR GENERATING MORE STEALTHY POISONING DATA

Fig. 2 demonstrates that the proposed DRS can effectively distinguish between clean and adversarial documents. This raises a question: if the attacker is aware that the defender will employ the DRS based detection, how will an attacker respond? In this section, we address this question by proposing new attack algorithms designed to generate more stealthy poisoning data that may bypass the previously established DRS scores.

The high-level idea behind this new series of attack algorithms is to apply regularization techniques when creating poisoning data to penalize large DRS scores for that data. Because each attack, as outlined in Table 1, has its own way of achieving the attacker's dual goals (as specified in Section 3), their corresponding new attack algorithms (by adding DRS regularization) may slightly differ depending on the context. In the following, we will use the attack proposed in Chen et al. (2024a) to demonstrate how our newly proposed attack algorithm works specifically.

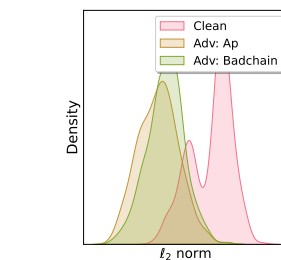 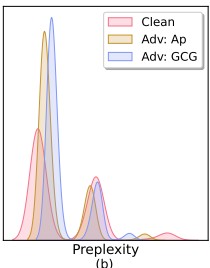 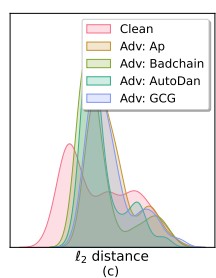 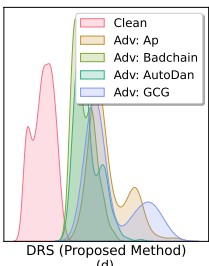

Figure 2: Density plot of different defense methods: (a) $\ell_2$ norms of clean and poisoned documents' embedding vectors; (b) perplexity of clean and poisoned documents; (c) $\ell_2$ distances to clean documents; (d) the proposed DRS scores for clean and poisoned documents. We observe that for existing defenses, namely Fig. (a), (b), and (c), the distributions of clean and poisoned documents under these defense mechanisms overlap significantly and cannot be separated. However, there is a sharp separation in the distribution of DRS between the clean documents and adversarial documents, indicating the effectiveness of DRS in detecting poisoned documents. For better visual clarity, in Fig. (a) and (b), we only show plots for certain attacks, but the overall conclusion remains the same.

The work Chen et al. (2024a) aims to find/generate effective red-teaming (i.e., poisoned) data in hopes of achieving the attackers' dual goals as outlined in Section 3. To be specific, they propose to minimize the following objective: $\min -\mathcal{O}_1 + \lambda_1 \mathcal{O}_2$, where $\mathcal{O}_1$ corresponds to the distance between clean and poisoned documents, $\mathcal{O}_2$ is the distance within poisoned documents, and $\lambda_1 > 0$ is a regularization parameter. Here, the variable to be optimized is the red-teaming (poisoned) data. Intuitively, their goal is to ensure that the poisoned data remain distinct from the clean data while minimizing the distances within the poisoned documents to enhance retrieval performance. However, their attacks can be effectively filtered out using our proposed DRS, as demonstrated in Fig. 2(d) under the attack name Ap. Now we propose to add a regularization term to the above original objective function, resulting in our proposed algorithm:

$$\min -\mathcal{O}_1 + \lambda_1 \mathcal{O}_2 + \lambda_2 \textbf{ DRS scores on poisoned data},$$

where the DRS scores on poisoned data are calculated according to Algorithm 1 and $\lambda_2 > 0$ is a regularizing parameter. Since the calculation of DRS does not involve any non-differentiable operations, existing gradient-based methods developed for optimizing the original objective functions (namely, the objectives without the DRS scores) will still remain effective. The resulting poisoned documents from this new objective will have smaller DRS scores compared to those generated under the original objective, making them more likely to bypass the proposed defense.

## 5 EXPERIMENTAL STUDIES

In this section, we conduct experiments across various setups to demonstrate the effectiveness of both our proposed attacks and defenses. On the defense side, we evaluate the proposed DRS against three different types of state-of-the-art attacks: (1) RAG attacks against Agent-LLMs (Chen et al., 2024a), (2) adversarial attacks on dense retrieval systems for general purpose Q&A (Long et al., 2024), and (3) knowledge poisoning attacks targeting medical Q&A RAG (Zou et al., 2024). We observe that the proposed DRS outperforms existing state-of-the-art techniques across all cases. On the attack side, we test our attack for the RAG Agent-LLM case (Chen et al., 2024a) to generate more stealthy red-teaming data. We observed a decreased detection rate of the red-teaming data generated by our attack compared to those generated directly from (Chen et al., 2024a). These results confirm the effectiveness of our proposed methods, supporting our theoretical findings. Due to space limitations, we will briefly describe the setups for each task in the main text and provide all the details in the appendix. All experiments were conducted on cloud computers equipped with Nvidia A100 GPUs.

### 5.1 THE EFFECTIVENESS OF PROPOSED DRS DEFENSE

#### 5.1.1 ATTACK I: RAG AGENT

**Autonomous Driver** Following the work of (Chen et al., 2024a), we consider the case of agents (e.g., autonomous drivers (Caesar et al., 2020)) equipped with LLMs that communicate using RAG systems. The attack goal is to generate red-teaming data that trick the agents into making

incorrect driving decisions while maintaining normal performance for clean queries. We employ four baseline methods for generating red-teaming data: Greedy Coordinate Gradient (GCG) (Zou et al., 2023), AutoDAN (Liu et al., 2023a), Corpus Poisoning Attack (CPA) (Zhong et al., 2023), and BadChain (Xiang et al., 2024). For each attack method, we generate 300 poisoned data samples. For the DRS parameters, we set $M$ (the dimensions to be calculated, as shown in Algorithm 1) to 100, the number of clean queries to 300 with $k = 5$, resulting in a total of 1,000 clean documents (after removing duplicates). We compare the detection performance of the poisoned data using three state-of-the-art defenses and our proposed defense. The results, shown in Table 2, indicate that our method significantly outperforms the others across all tasks, and the proposed DRS often achieves near-perfect accuracy.

**Re-Act Q&A** Following the work of (Chen et al., 2024a), we consider the case of the ReAct agent Q&A (Yao et al., 2022). All other setups remain the same as above. The results, shown in the third row of Table 3, indicate that our method significantly outperforms the others across all tasks, with the proposed DRS often achieving near-perfect accuracy.

Table 2: Filtering rates (↑ better) for poisoned data (in the RAG agent context (Chen et al., 2024a)), generated by four attacks across two tasks and evaluated with four different defenses. The decision threshold for filtering is set to the 99th percentile of the *clean* scores, resulting in a false positive rate of approximately 1% for clean documents.

| Task | Attack | Defense | | | |
|---|---|---|---|---|---|
| | | Perplexity filter | $\ell_2$-norm filter | $\ell_2$-distance filter | DRS (proposed) |
| | AgnetPoison | 0.03 | 0.02 | 0.01 | 0.99 |
| Agent-Driver | BadChain | 0.03 | 0.03 | 0.01 | 0.99 |
| | AutoDan | 0.02 | 0.10 | 0.01 | 0.99 |
| | GCG | 0.03 | 0.01 | 0.02 | 0.99 |
| | AgnetPoison | 0.01 | 0.34 | 0.03 | 0.99 |
| ReAct-StrategyQA | BadChain | 0.01 | 0.02 | 0.01 | 0.99 |
| | AutoDan | 0.11 | 0.01 | 0.06 | 0.99 |
| | GCG | 0.01 | 0.01 | 0.01 | 0.99 |

### 5.1.2 ATTACK II: DENSE PASSAGE RETRIEVAL FOR GENERAL PURPOSE Q&A

We follow the setup of the work by (Long et al., 2024), which proposed backdoor attacks for dense passage retrievers used in general-purpose Q&A systems. We report the results under different backdoor/poisoning ratio in Table 3. First, we observed a decrease in the detection rate of our proposed method, although it remains significantly higher than all other state-of-the-art methods. One potential reason for the decreased filtering rate is that (Long et al., 2024) introduced poisoned documents by only incorporating simple grammar errors, such as subject-verb agreement mistakes (`She go to the store` instead of `She goes to the store`). As a result, the poisoned documents are not sufficiently abnormal compared to their clean versions, which is further evidenced by the low attack success rate of their attacks compared to the agent attacks discussed previously.

### 5.1.3 ATTACK IV: KNOWLEDGE POISONING FOR MEDICAL Q&A RAG

We follow the setup of RAG for medical Q&A (Xiong et al., 2024) and employ PosionedRAG (Zou et al., 2024) for generating poisoned documents. Some details are listed as follows. **Query** Following (Xiong et al., 2024), we use a total of three sets of queries, including three medical examination

Table 3: Filtering rates (↑ better) for poisoned data (in the dense retrieval context for general domain Q&A), generated by BadDPR (Long et al., 2024) and evaluated with four different defenses. The decision threshold for filtering is set to the 99th percentile of the *clean* scores, resulting in a false positive rate of approximately 1% for clean documents.

| Backdoor Ratio | Perplexity filter | $\ell_2$-norm filter | $\ell_2$-distance filter | DRS (proposed) |
|---|---|---|---|---|
| 1% | 0.03 | 0.02 | 0.01 | 0.49 |
| 5% | 0.02 | 0.04 | 0.05 | 0.50 |
| 10% | 0.18 | 0.27 | 0.25 | 0.57 |
| 20% | 0.13 | 0.36 | 0.36 | 0.65 |

QA datasets: MedQAUS, MedMCQA, and PubMedQA. **Medical Corpus** Following (Xiong et al., 2024), we select a total of two medical-related corpora: (1) Textbook (Jin et al., 2021) ($\sim 126\text{K}$ documents), containing medical-specific knowledge, and (2) PubMed, which consists of biomedical abstracts. **Retriever** We select two representative dense retrievers: (1) a general-domain semantic retriever: Contriever (Izacard et al., 2021), and (2) a biomedical-domain retriever: MedCPT (Jin et al., 2023). We summarize the results for the attack described in (Zou et al., 2024), using Contriever as the retriever and the textbook as the corpus, in Table 4 below. We observed that our method significantly outperforms the others.

Table 4: Filtering rates ($\uparrow$ better) for poisoned data (in the context of Medical Q&A), generated by PoisonedRAG attack (Zou et al., 2024). The decision threshold for filtering is set to the 99th percentile of the *clean* scores, resulting in a false positive rate of approximately $1\%$ for clean documents.

| Retriever | Task | Defense | | | |
|---|---|---|---|---|---|
| | | Perplexity filter | $\ell_2$-norm filter | $\ell_2$-distance filter | DRS (proposed) |
| Contriever | MedQAUS | 0.01 | 0.80 | 0.02 | 0.96 |
| | MedMCQA | 0.08 | 0.90 | 0.23 | 0.96 |
| | PubMedQA | 0.07 | 0.81 | 0.11 | 0.95 |
| MedCPT | MedQAUS | 0.01 | 0.61 | 0.03 | 0.96 |
| | MedMCQA | 0.08 | 0.52 | 0.04 | 0.96 |
| | PubMedQA | 0.07 | 0.41 | 0.12 | 0.95 |

## 5.2 THE EFFECTIVENSS OF PROPOSED ATTACKS

In this section, we test our proposed attacking algorithm to demonstrate that the previously developed DRS can be less effective at detecting poisoning data generated by our algorithm. As described in Section 4.2, we introduce a regularization term into the original AgentPoison attack formulation to penalize large DRS scores for the poisoned data. For the hyperparameter $\lambda_2$, which controls the strength of the regularization, we select a value such that the attack success rate of the poisoned data remains comparable to that generated by the original AgentPoison.

The results for the Agent-Driver task are summarized in Table 5. We observe that the DRS detection rate for poisoning data generated by our proposed algorithm decreases by $15\%$, highlighting the effectiveness of the algorithm. Furthermore, the DRS detection rate can be reduced further by increasing the penalty magnitude $\lambda_2$. However, this comes with a trade-off: as the penalty increases, the attack success rate of the corresponding poisoned data decreases, as predicted by our theorems. Additional ablation studies can be found in the appendix.

Table 5: Filtering rates ($\uparrow$ better) for poisoned data, generated by AgentPoison and our newly proposed DRS-regularized AgentPoison. The decision threshold for filtering is set to the 99th percentile of the *clean* scores, resulting in a false positive rate of approximately $1\%$ for clean documents.

| Attack Method | Perplexity filter | $\ell_2$-norm filter | $\ell_2$-distance filter | DRS (proposed) |
|---|---|---|---|---|
| AgentPoison | 0.03 | 0.03 | 0.01 | 0.99 |
| DRS-regularized AgentPoison | 0.03 | 0.01 | 0.01 | 0.85 |

## 6 CONCLUSION

In this work, we study the safety issues associated with using RAG. In particular, we first show that most existing RAG attacks are essentially targeted data poisoning attacks. We then provide a unified framework to examine these attacks and offer insights into their effectiveness. Specifically, we demonstrate that more effective poisoning attacks tend to occur in directions where the clean data distribution exhibits low variance. Based on these insights, we propose a new defense for detecting poisoned data and introduce a series of new attacking algorithms that can potentially lead to more stealthy (in terms of detection) data. We test both our proposed attacks and defenses on various applications and observe consistent improvements. Proofs, detailed experimental setups, and additional ablation studies are included in the appendix.

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
