# OpenReview forum: "Understanding Data Poisoning Attacks for RAG: Insights and Algorithms"
_ICLR.cc/2025/Conference — Submitted to ICLR 2025_

### Official Review · Reviewer_NLx3 · 2024-10-28

**Soundness:** 2
**Presentation:** 2
**Contribution:** 2
**Rating:** 5
**Confidence:** 3

**Summary:**

In this paper, the authors conduct a comprehensive analysis of data poisoning attacks on RAG. Specifically, they provide a framework to analyze attacker objectives. They observe that more effective attacks tend to result in larger relative shifts along directions with smaller variances. Based on this observation, the authors design a new filtering method to defend against poisoning attacks. Additionally, they introduce a regularizer to bypass the new detection method. Through experiments, they demonstrate the effectiveness of both the new defense and attack strategies.

**Strengths:**

The analysis and observations of current poisoning attacks on RAG are novel and interesting.

The paper considers four attack settings to demonstrate the effectiveness of the defense methods, offering a comprehensive and thorough evaluation.

**Weaknesses:**

Major concern: I am uncertain about the reliability of DRS. For example, if the question is, "Who is the OpenAI CEO?" I would expect the embedding of a clean document ("The CEO of OpenAI is Sam Altman") to be similar to that of a poisoned document ("The CEO of OpenAI is Elon Musk"). I am unsure whether DRS can effectively handle such an attack.

The clarity of this paper needs improvement.
Some examples:
1. In Figure 1, what is the Y-axis?
2. In Section 2.1, the attacker’s capability is described as "only injecting poisoned data (e.g., by creating a new Wikipedia page)." However, in Section 5.1.2, the setting appears to change, with the retriever itself being backdoored.
3. In Section 5.1.1, there is no description of the adversarial query.
4. In Section 5.1.1, the statement "For each attack method, we generate 300 poisoned data samples" is unclear. Does "poisoned data samples" refer to poisoned documents?

If I understand correctly, DRS also requires a set of clean samples to compute the threshold, but it is unclear how large and diverse this dataset needs to be.

**Questions:**

NA

---

> ### Author Response · Authors · 2024-11-21
> **Rebuttal**
>
> We deeply appreciate the reviewers for dedicating their time and effort to reviewing our manuscript and providing insightful feedback. We are pleased that all reviewers acknowledged the novelty of our work. Furthermore, we are grateful that they considered our writing clear and our approach effective across different setups. We will integrate their suggestions into the revised version of the paper.
>
> >Q: I am uncertain about the reliability of DRS. For example, if the question is, "Who is the OpenAI CEO?" I would expect the embedding of a clean document ("The CEO of OpenAI is Sam Altman") to be similar ...
>
> **Response**: Thank you for your insightful and sharp question regarding the reliability of DRS. We believe that, in general, it could be challenging to distinguish between the two queries you provided using all existing defense methods, including ours. These queries are very close in terms of semantic meaning, sentence quality, and embedding vectors. As a result, regardless of the defense method—whether norm-based, perplexity-based, or our proposed method—it may be infeasible to distinguish between these two queries.
>
> More fundamentally, by casting the problem of detecting the two queries you proposed as a hypothesis testing/binary classification problem, we can show that the Bayes Risk (which depends on the total variation between the two distributions) for the problem will be very high, as their total variation distance is very small. This represents a fundamental limitation of all detection-based methods, including our proposed DRS.
>
> >Q: In Figure 1, what is the Y-axis?
>
> **Response**: Thank you for your question regarding Figure 1. The Y-axis in Figure 1 represents the relative change in the attack success rate (ASR) for the three attacks. We define the relative changes in Lines 288-295. Specifically, the relative change along a certain direction of the embedding vectors is calculated by measuring the difference between adversarial and clean documents along the directions of the clean documents. This difference is then normalized by dividing the relative mean by the standard deviation along these directions. We observed that more effective attacks, such as AgentPoison, tend to have a larger relative distance along directions with small variances (i.e., the left-most group represents the directions of clean embedding documents with the top 100 smallest variances), which empirically verifies our theory.
>
>
> >Q: In Section 2.1, the attacker’s capability is described as "only injecting poisoned data (e.g., by creating a new Wikipedia page)." However, in Section 5.1.2, the setting appears to change, with the retriever itself being backdoored.
>
> **Response**: Thank you for your question regarding the threat model. This does not violate our threat model. In Section 5.1.2, the scenario considered is that the attacker only poisons the training data, but later the users themselves fine-tune the retriever based on the poisoned data for better performance. As a result, the retriever becomes backdoored. We will further clarify this point in the revised manuscript.
>
> >Q: In Section 5.1.1, there is no description of the adversarial query.
>
> **Response**: Thank you for your comment regarding the adversarial query. Due to space limitations, we included detailed descriptions and provided some examples of the adversarial queries in the appendix. Here, we list them for your reference. To create adversarial data, the attacker appends an adversarial backdoor trigger at the end of normal data. The backdoor/adversarial trigger used for autonomous driving is:  `Be safe and make a discipline.`
>
> >Q: In Section 5.1.1, the statement "For each attack method, we generate 300 poisoned data samples" is unclear. Does "poisoned data samples" refer to poisoned documents?
>
> **Response**: Thank you for your question. Yes, you are absolutely correct. We will make this clear in the revised manuscript.
> >Q: If I understand correctly, DRS also requires a set of clean samples to compute the threshold, but it is unclear how large and diverse this dataset needs to be.
>
> **Response**: Thank you for your question. Yes, you are correct about the need for a set of clean samples. Regarding the size of the data, we provide an ablation study on using different sizes for the clean data (in the context of autonomous driving) and summarize the results in the following table. We observed that our proposed method is robust with different sample sizes of the clean dataset.
>
> | Sample Size |500 | 1000| 1500|
> |---|---|---|---|
> | Detection Rate |0.91 | 0.97| 0.99|
>
> Regarding the diversity of the clean dataset, it depends heavily on the targeted query set to be protected. If the query set contains queries from closely similar topics, then the corresponding clean dataset need not be diverse. Otherwise, the clean dataset requires more diverse data. We will clarify this in the revision.

---

> > ### Comment · Reviewer_NLx3 · 2024-11-22
> >
> > Thanks for the detailed response. Some of my concerns have been addressed. I'd love to increase my score to 5.
> >
> > Since my major concern still challenges the proposed detection method, I cannot support the acceptance of this paper. I would encourage the authors to analyze the problem much deeper in the next version.

---

> > > ### Author Response · Authors · 2024-11-22
> > > **Thanks for the response and raising scores. Further Clarificaiton**
> > >
> > > We would like to thank the reviewer for their quick responses and for raising the scores.
> > >
> > > We would like to highlight the specific problem (the detection challenge) that the reviewer is mentioning. The scenario described by the reviewer is a type of **fundamentally infeasible** problem (the hardness of which we prove in the following). In the sense that **no detectors** can perform well under such a case.
> > >
> > > We believe it may not be reasonable to focus on this scenario particularly, especially given that **our method has shown significant improvements** over the baselines in various experimental setups (e.g., increasing the detection rate from 0.1 to 0.99).
> > >
> > > Theoretically, let us denote the distributions of the two queries you proposed as $P_1$ and $P_2$. The Bayes Risk (which is the best possible performance for any classifier) is:
> > >
> > > $$
> > > \frac{1}{2}[1 - TV(P_1, P_2)],
> > > $$
> > >
> > > where $TV$ is the total variation distance between the two distributions. In the scenario you mentioned, the distributions of $P_1$ and $P_2$ are very close to each other, and hence the Bayes Risk is very close to 0.5 (which is the performance of a random guess). In other words, **the optimal strategy (for any detection) is to randomly guess**. This is a fundamental limitation of the problem itself, regardless of any detection methods.
> > >
> > > We hope that the reviewer can take this part into account. Thank you again for your effort and time.

---

> > > > ### Comment · Reviewer_NLx3 · 2024-11-27
> > > >
> > > > Thank you again for the detailed response.
> > > >
> > > > I understand that the scenario might be fundamentally infeasible. However, constructing such poisoning samples should be extremely easy (much easier than methods like GCG).

---

> > > > > ### Author Response · Authors · 2024-11-27
> > > > > **Further responses**
> > > > >
> > > > > Thank you for your further responses.
> > > > >
> > > > > We are pleased that you also agree with the potential impossibility detection result of the scenario you mentioned. While it is true that crafting poisoning examples in this context can be easy, **attacks under this regime do NOT necessarily guarantee high success rates**. In fact, we can show that the attack success rates will be low if the distributions of clean and poisoned data are very similar. This is because when the distributions of adversarial/poisoned and clean data are close, both types may be simultaneously retrieved by a relevant query. Such behavior contradicts the underlying philosophy of all (as well as our) RAG poisoning attacks, as outlined in our threat model, where the goal is to ensure that all retrieved documents are poisoned given adversarial queries. As a result, attacks of this type may not be favored in practice, since the fundamental requirement of high attack success rates is not met.
> > > > >
> > > > > We will use the example you proposed as a demonstration. When the adversary query "Who is the CEO of OPENAI?" is made, both the clean ("The CEO of OpenAI is Sam Altman") and poisoned ("The CEO of OpenAI is Elon Musk") documents are retrieved, as we empirically verified using the WikiQA dataset and the Contriever embedding function, following the PoisonedRAG approach. This retrieval performance actually indicates the failure of the attack, since clean documents are also retrieved, which contradicts the adversary's goal. When the LLM is provided with both clean and poisoned documents, it is likely that it will not output the adversary's targeted answer, leading to a low attack success rate (ASR). We found that the ASR in such cases is around 40%, further indicating the failure of such attacks.
> > > > >
> > > > > To summarize, from a detection perspective, your insightful examples fall into the category of impossible results, making all methods ineffective. **However, such attacks are unlikely to be of practical interest, as they are expected to have a low success rate, which contradicts the attacker's goal. As a result, while such cases theoretically exist, they are unlikely to have a significant impact on real implementations. Overall, we believe these scenarios will not negatively affect our proposed defense.**
> > > > >
> > > > > We hope that the reviewer can take this part into account. Thank you again for your effort and time.

---

### Official Review · Reviewer_F894 · 2024-11-02

**Soundness:** 2
**Presentation:** 2
**Contribution:** 2
**Rating:** 5
**Confidence:** 4

**Summary:**

This paper investigates vulnerabilities in RAG systems due to adversarial data poisoning attacks. The authors analyze how specific data characteristics affect attack success, proposing a new defense method, Directional Relative Shifts (DRS), which detects poisoned data by monitoring shifts in directions with low data variance. They also introduce a stealthier attack algorithm that minimizes DRS to evade detection. Experimental results indicate that DRS demonstrates strong defense performance, though its effectiveness is somewhat reduced against the proposed attacks.

**Strengths:**

1. **Innovative Approach** -- The proposed DRS defense is novel in its focus on low-variance directions to detect adversarial data shifts. This approach, within the experimental settings of the paper, demonstrates defensive effectiveness against poisoning attacks.
2. **Comprehensive Evaluation** -- This paper provides extensive experiments in multiple RAG setups, such as autonomous driving and medical Q&A, confirming the generalizability of DRS across diverse applications.
3. **Insightful Theoretical Contributions** -- The theoretical analysis connecting attack effectiveness to data distribution characteristics (specifically low-variance directions) offers valuable insights, potentially influencing future defenses in retrieval systems.

**Weaknesses:**

1. **Sparse Theoretical Explanation** -- While DRS’s foundation on variance shifts is intuitive, a deeper theoretical analysis could further clarify why certain dimensional shifts are more vulnerable. This would strengthen the defense’s theoretical underpinnings.
2. **Unrealistic Defense Assumptions** -- The defense method assumes prior knowledge of a specific subset of queries that need protection from poisoning attacks. In real-world applications, defenders typically do not have knowledge of which specific queries might be targeted, and a practical defense would need to offer broad protection across all possible queries. This limitation reduces the generalizability and practicality of the proposed DRS-based defense method.
3. **Unrealistic Assumption** -- In Section 3.1, the authors illustrate their attack method with an example where, in a knowledge base about food, an adversarial query about mathematics is used to avoid retrieving clean documents. This assumption is unrealistic, as it does not reflect typical user behavior—users are unlikely to ask irrelevant questions, like mathematics queries, in a food-related knowledge base context. This reduces the practical applicability of the assumptions underpinning the theoretical insights.
4. **Inaccurate Description of Experimental Results** -- In Figure 1, the authors claim that "we can observe that the attack success rates of Ap are higher than BadChain and AutoDan." However, the figure only shows relative changes in certain dimensions and does not explicitly provide data on the actual success rates of each attack. This discrepancy between the description and the figure may mislead readers and reflect a lack of rigor in interpreting experimental results.
5. **Limited Innovation in Attack Method** -- Although the paper claims to develop a new attack algorithm, it essentially modifies existing attack methods by adding a regularization term based on the proposed defense metric (DRS). This adjustment is an incremental improvement rather than a substantive innovation. Moreover, the effectiveness of this “new” attack is limited, as it only partially reduces the DRS defense success rate without significantly overcoming the defense.

**Questions:**

1. **Clarification on Theoretical Basis** -- Could you provide a more rigorous theoretical explanation for why certain low-variance directions are more susceptible to poisoning attacks in DRS? A deeper analysis would help clarify the underlying vulnerabilities exploited by attackers.
2. **Defense Scope and Practicality** -- Given that the defense currently focuses on protecting a specific subset of pre-selected queries, how would DRS perform in scenarios where the entire query space needs protection? Have you considered evaluating DRS’s effectiveness without pre-selecting queries, to simulate more realistic defensive conditions?
3. **Lack of Attack Success Rate Comparison** -- In the evaluation of the proposed “new” attack algorithm, the paper only presents its detection rate under the DRS defense. Could you provide a comparison of the attack success rates between the new algorithm and traditional attacks?

---

> ### Author Response · Authors · 2024-11-21
> **Rebuttal**
>
> We deeply appreciate the reviewers for dedicating their time and effort to reviewing our manuscript and providing insightful feedback. We are pleased that all reviewers acknowledged the novelty of our work. Furthermore, we are grateful that they considered our writing clear and our approach effective across different setups. We will integrate their suggestions into the revised version of the paper.
>
> >Q: Sparse Theoretical Explanation -- While DRS’s foundation on variance shifts is intuitive, a deeper theoretical analysis could further clarify why certain dimensional shifts are more vulnerable. This would strengthen the defense’s theoretical underpinnings.
>
> **Response**: Thank you for your suggestion regarding the theoretical explanation of DRS. In Corollary 1, we have shown that certain directions are more effective for attacks, with an intuitive explanation provided in Remark 3. Below is the logic flow of our theoretical analysis as it was presented for your reference:
>
> 1. Theorem 1 demonstrates that attackers can successfully launch an attack, as defined in Section 2.1, by creating poisoned queries sufficiently distant from normal ones.
> 2. However, if adversarial queries are too different from the clean ones, they may be easily detected. Therefore, the attacker seeks to answer: **Given the maximum deviation (e.g., ℓ2 distance) between the normal and adversarial distributions, what are the most effective directions for moving from $Q_{normal}$ to $Q_{adv}$?**
> 3. Corollary 1 shows that the most effective attack directions are those that maximize the rate at which the clean data $\mathcal{D}^{\text{clean}}$ density decays.
>
>
> Intuitively, directions in $\mathcal{D}^{\text{clean}}$ with rapidly decaying density often correspond to low-variance directions. Low variance indicates that most of the probability mass is concentrated around the mean, so even small deviations from the mean significantly reduce the probability mass. This aligns with the attacker's goal: perturbing a clean query in a low-variance direction reduces the likelihood of clean data being close to the perturbed query, increasing the chance of retrieving poisoned documents. We hope this explanation clarifies the theoretical basis of DRS and will improve the presentation in the revised manuscript.
>
>
> >Q: Unrealistic Defense Assumptions -- The defense method assumes prior knowledge of a specific subset of queries that need protection from poisoning attacks. In real-world applications, defenders typically do not have knowledge of which specific queries might be targeted, and a practical defense would need to offer broad protection across all possible queries. This limitation reduces the generalizability and practicality of the proposed DRS-based defense method.
>
> **Response**: Thank you for your comment regarding the defense assumption. We believe that the assumption of prior knowledge of a specific subset of queries is reasonable and, in fact, essential for both practical and theoretical reasons.
>
> 1. Theoretically, if the defender has no prior knowledge of the queries to be protected or aims to protect all possible queries, it can be shown that (i.e., using LeCam's Method to prove an information-theoretical lower bound) the defense is infeasible. This is because, by considering all possible queries, their distributions are likely to cover the entire input space, making it impossible to distinguish between normal and adversarial queries.
>
> 2. Practically, in many real-world applications, the defender does have some knowledge of the queries that need protection. For example, a defender, such as a RAG service provider, has access to the underlying database for retrieval and is likely aware of which queries are critical to the system's operation. In this case, the defender can use this knowledge to protect these queries from poisoning attacks.
>
> Moreover, the effectiveness of the proposed defenses across multiple settings clearly demonstrates the wide applicability of the defense. We will further clarify this point in the revised manuscript.
>
> >Q: Unrealistic Assumption -- In Section 3.1, the authors illustrate their attack method with an example where, in a knowledge base about food, an adversarial query about mathematics is used to avoid retrieving clean documents. This assumption is unrealistic, as it does not reflect typical user behavior ...
>
> **Response**: Thank you for your comments regarding the experimental results. We believe there might be a misunderstanding between the Assumption 1 (we assume you are referring to this one) and the example. We provided this extreme-case example simply to offer some sanity checks/intuition. Recall that when proving theoretical results, it is common to choose extreme values for a simple sanity check. We will modify the example to make it more realistic in the revised manuscript.

---

> > ### Author Response · Authors · 2024-11-21
> > **Rebuttal continues**
> >
> > >Q: Inaccurate Description of Experimental Results -- In Figure 1, the authors claim that "we can observe that the attack success rates of Ap are higher than BadChain and AutoDan." However, the figure only shows relative changes in certain dimensions and does not explicitly provide data on the actual success rates of each attack. This discrepancy between the description and the figure may mislead readers and reflect a lack of rigor in interpreting experimental results.
> >
> > **Response**: Thank you for your comments regarding the experimental results. The ASR of the attacks mentioned is AP > BadChain ≥ AutoDan (we have included this information in the caption of Figure 1). We will revise the figure and the corresponding text to make it clearer in the revised manuscript.
> >
> > >Q: Limited Innovation in Attack Method -- Although the paper claims to develop a new attack algorithm, it essentially modifies existing attack methods by adding a regularization term based on the proposed defense metric (DRS). This adjustment is an incremental improvement rather than a substantive innovation. Moreover, the effectiveness of this “new” attack is limited, as it only partially reduces the DRS defense success rate without significantly overcoming the defense.
> >
> > **Response**: Thank you for your comments regarding the attacks. To the best of our knowledge, there is very little existing work, as cited in the paper (as of the date of submission), that proposes similar attack algorithms aimed at reducing stealthiness in data poisoning for RAG systems. Our method is novel in that it specifically targets this challenge of minimizing the detectability of attacks in the context of RAG systems. However, if you are aware of any relevant work, please let us know, and we would be more than happy to cite it.
> >
> > In addition, the main contribution of our paper is threefold: (1) a novel theoretical understanding of poisoning attacks in RAG systems, (2) a new defense mechanism based on these insights, and (3) a new attack algorithm designed to challenge this defense. We believe these significant contributions highlight the overall novelty of our work.
> >
> >
> > >Q: Lack of Attack Success Rate Comparison -- In the evaluation of the proposed “new” attack algorithm, the paper only presents its detection rate under the DRS defense. Could you provide a comparison of the attack success rates between the new algorithm and traditional attacks?
> >
> > **Response**: Thank you for your comments regarding the experimental results. We report the attack success rates of the proposed DRS attack and the traditional attacks (i.e., AP) in the table below. We observed that the proposed DRS attack has a similar attack success rate to the traditional attack, indicating that the proposed attack is effective in generating poisoned data while maintaining stealthiness. We will include this table in the revised manuscript.
> >
> > **Table**: Attack success rates of the proposed DRS attack and the traditional (AP) on the autonomous driving task.
> >
> > |Task | Metric                    | AP | DRS (proposed) |
> > |----|-----------------------------------|----|----------------|
> > |Autonmous Driving | Attack Success Rate               | 0.81 | 0.78          |
> > |ReAct | Attack Success Rate               | 0.73 | 0.74        |

---

> ### Comment · Reviewer_F894 · 2024-11-26
>
> Thanks for the responses. Since the responses have addressed part of my concern, I will increase my score.

---

> > ### Author Response · Authors · 2024-11-26
> > **Thanks for your feedback and increasing the score!**
> >
> > Thank you for your feedback and for raising the score! We are happy that our responses have addressed your concerns.
> >
> > Thank you again for your effort in reviewing our paper!

---

### Official Review · Reviewer_bSEM · 2024-11-04

**Soundness:** 3
**Presentation:** 3
**Contribution:** 2
**Rating:** 6
**Confidence:** 3

**Summary:**

The paper investigates the vulnerability of Retrieval-Augmented Generation (RAG) systems to data poisoning attacks, where adversaries manipulate the retrieval corpus to influence model outputs. It reveals that effective poisoning occurs along low-variance directions in the clean data distribution, allowing attackers to insert poisoned data that stealthily alters retrieval results. The authors propose a new defense metric, Directional Relative Shifts (DRS), to detect these poisoned entries by examining shifts along susceptible directions. Additionally, they introduce an advanced attack algorithm that regularizes DRS values, making poisoned data harder to detect. Empirical tests confirm the effectiveness of DRS in various RAG applications, demonstrating the need for robust defenses.

**Strengths:**

1.	The authors attempt to give a deeper understanding and theoretical analysis of existing attacks. It should be encouraged.
2.	This is a well-written paper. The definitions of symbols and the overall flow are clear.
3.	The proposed defense is simple yet highly effective.

**Weaknesses:**

1. Missing some references.
- Line 65: The authors should provide references for perplexity-based filters (e.g., [1]).
- Line 143-153: The authors should also mention existing attacks against (e.g., [2]).
2. There has been some work discussing the characterization of poisoned samples. In particular, the proposed method (i.e., DRS) is similar to [3] to some extent. The authors should compare their method to existing works.
3. The authors only use AgentPoison as an example to demonstrate the effectiveness of the proposed attack. The authors should conduct more extensive experiments on all discussed attacks to verify its generalizability.
4. According to Section 5.2 (Table 5), the performance of the proposed attack is limited.
5. The authors should directly place the appendix after the references in the main document.


References
1. Onion: A Simple and Effective Defense against Textual Backdoor Attacks.
2. Targeted attack for deep hashing based retrieval.
3. Spectral Signatures in Backdoor Attacks.

**Questions:**

1. Add more related references.
2. compare their method to existing works like [3].
3. Conduct more experiments regarding the proposed attacks.
4. Explain the performance of the proposed attack.

Please find more details in the aforementioned 'Weaknesses' part.

PS: I am willing to increase my score if the authors can (partly) address my concerns.

---

> ### Author Response · Authors · 2024-11-21
> **Rebuttal**
>
> We deeply appreciate the reviewers for dedicating their time and effort to reviewing our manuscript and providing insightful feedback. We are pleased that all reviewers acknowledged the novelty of our work. Furthermore, we are grateful that they considered our writing clear and our approach effective across different setups. We will integrate their suggestions into the revised version of the paper.
>
> >Q: Missing some references.
>
> **Response**: Thank you for pointing out the missing references. We have included them in the revised manuscript.
>
> >Q: There has been some work discussing the characterization of poisoned samples. In particular, the proposed method (i.e., DRS) is similar to [3] to some extent. The authors should compare their method to existing works.
>
> **Response**: Thank you for your valuable suggestion regarding the comparison of our proposed method to existing works. The proposed DRS differs significantly from [3] in terms of the threat model and technical components.
>
> [3] is a **training-stage defense** method that uses **both** clean and poisoned training data to filter out backdoors and build a clean model, evaluating based on clean data and attack success rates. In contrast, our approach, being **an inference-stage** method, uses only a small set of clean validation data and no knowledge of the poisoned data, aiming to detect future poisoned inputs. We evaluate our method using the AUC-ROC score of the detector.
>
>
>
> >Q: Explain the performance of the proposed attack.
>
> **Response**: Thank you for your question regarding the performance of the proposed attack. In Table 5 in Section 5.2 of the main text, we report the filtering rates for poisoned data generated by different algorithms. Given the same attack success rates of the poisoned data, more effective attacks have lower filtering rates, as they are less likely to be detected by the defense mechanism. We observe that the detection rate (by the proposed DRS defense) for poisoned data generated by our algorithm decreases by 15% compared to the vanilla AgentPoison, highlighting the effectiveness of the algorithm.
>
> In fact, as discussed in Section 4.2 of the main text, the detection rate of the proposed DRS defense can be further lowered by increasing the hyperparameter $\lambda_2$. However, this comes with a trade-off: as the penalty increases, the attack success rate of the corresponding poisoned data decreases, as suggested by our theorems. This is because, intuitively, a large penalty forces the poisoned data to be more similar to the clean data, making the attack ineffective.
>
>
>
> >Q: The authors only use AgentPoison as an example to demonstrate the effectiveness of the proposed attack. The authors should conduct more extensive experiments on all discussed attacks to verify its generalizability.
>
> **Response**: Thank you for your valuable suggestion regarding the evaluation of our newly proposed attack. To the best of our knowledge, we are not aware of any other existing attacks (with open-sourced code) that aim to achieve similar goals, i.e., generating poisoned data with high attack success rates while explicitly forcing them to be less discernible to detection, in the context of RAG, except for the ones already included. In fact, the AgentPoison attack (first on arXiv in July this year) itself is very new, as quoted: '... the first backdoor attack targeting generic and RAG-based LLMs ...'.
>
> Nonetheless, we have introduced a new attack (proposed by ourselves, motivated by backdoor literature) that penalizes the Wasserstein distance between adversarial and normal queries, instead of the proposed DRS. We summarize the results in the following table, where we observe that the detection rate of the Wasserstein distance-based attack is higher than that of the proposed DRS-based attack (with a lower detection rate indicating that the attack is more effective), indicating the effectiveness of our proposed DRS attack algorithm.
>
> **Table**: Filtering rates for poisoned data, generated by AgentPoison and our newly proposed DRS-regularized AgentPoison, and the Wasserstein-regularized AgentPoison. The decision threshold for filtering is set to the 99th percentile of the **clean** scores, resulting in a false positive rate of approximately 1% for clean documents.
>
> | Attack Method                     | Perplexity filter | ℓ2-norm filter | ℓ2-distance filter | DRS (proposed) |
> |-----------------------------------|-------------------|----------------|-------------------|----------------|
> | AgentPoison                       | 0.03              | 0.03           | 0.01              | 0.99           |
> | DRS-regularized AgentPoison       | 0.03              | 0.01           | 0.01              | 0.85           |
> | Wass-regularized AgentPoison (New)| 0.03              | 0.02           | 0.01              | 0.94          |

---

> > ### Comment · Reviewer_bSEM · 2024-11-23
> >
> > I would like to thank the authors for the detailed responses. However, some of my concerns remain.
> >
> > 1. Please compare to [3] from a technical aspect, instead of simply the aspect of threat model.
> > 2. A 14% reduction isn't enough for your attack to escape your detection algorithm.
> > 3. I look forward to seeing different types of attacks rather than just replacing distance metric.
> > - There are currently at least three related work for poisoning RAG that I am aware of. The absence of open source code is not a reason for you not to compare. Targeting only one piece of work does not constitute so-called 'understanding' and will also greatly limit the scope of your paper.
> > - I'm curious as to why not just take inspiration from existing backdoor attacks (e.g., different trigger designs)?

---

> > > ### Author Response · Authors · 2024-11-23
> > > **Thanks the reviewer for the feedback; Further Clarification**
> > >
> > > We thank the reviewer for the feedback. We would like to address the remaining concerns in the following points:
> > >
> > > 1. First, the difference in the threat model compared with [3] plays a significant role in the technical contribution of our work. In short, our technique primarily relies on the **localization property of the KNN algorithm**, rather than **robust statistics techniques** as those employed in [3]. To be more specific, in [3], the authors have access to both **clean and backdoor data**, and their technique involves applying **robust statistics** to eventaully build a classifier that is robust against the backdoor training data. However, in our setting, we **do not have access to the backdoor data** and only have access to clean data. Our theoretical contribution is to demonstrate how optimal backdoor data might look **even without direct access to them**. In this sense, our work addresses a potentially more difficult problem than the one in [3], as we have less information (i.e., no access to backdoor data). We derive our theory based on the **localization property of the KNN algorithm** and the **decaying property of distributions with well-behaved tails**.
> > >
> > > 2. Second, we can further lower the detection rate by adjusting the hyperparameter $\lambda_2$. For example, by increasing $\lambda_2$ to 0.75, we can lower the detection rate from 0.99 to 0.71, although the attack success rate would decrease by around 12%. We believe that, given the relatively recent development of this field, the reduction of 15% reported in the paper represents a significant improvement.
> > >
> > > 3. Third, while we are aware of several RAG poisoning attacks, as summarized in Table 1, **we are not aware of any attacks that explicitly aim to make generated poisoned data less discernible to detection by using specific objective functions/regularization techniqes in RAG setup**, except for the one included in our paper. If the reviewer is aware of other such works, we kindly request that they share them with us.
> > >
> > > 4. Fourth, we have conducted a new study (attack) in which we apply our technique to the Backdoor DPR Attack setting (Long et al., 2024). Specifically, we first trained a backdoored retriever based on backdoor data, where the backdoor triggers are primarily grammar errors. Next, we further optimize these backdoor triggers with the proposed DPR. The results are summarized in the table below, where we observe a decrease in the detection rate from 0.65 to 0.52. We believe the detection rate can be further reduced by tuning the hyperparameters.
> > >
> > > | Attack Method                   | Perplexity Filter | ℓ2-norm Filter | ℓ2-distance Filter | DRS (Proposed) |
> > > |----------------------------------|-------------------|----------------|--------------------|----------------|
> > > | BadDPR                           | 0.13              | 0.36           | 0.36               | 0.65           |
> > > | DRS-regularized BadDPR      | 0.10              | 0.31           | 0.37               | 0.52           |
> > >
> > > 5. Finally, we note that NLP backdoors are much less diverse than CV backdoors due to the discrete nature of text. In other words, the backdoor trigger or pattern is quite restricted, typically involving just a few words or sentences. In this context, we believe that the AgentPoison paper (accepted at NeurIPS this year) and the newly added backdoor DPR scenario are indeed representative of the current state of the art.
> > >
> > > We hope that these points address the reviewer's concerns. We are happy to provide further clarification if needed. Thank you again for your time and effort in reviewing our work!

---

> > > > ### Comment · Reviewer_bSEM · 2024-11-24
> > > >
> > > > Thank you for your detailed responses! Your rebuttal has addressed most of my concerns. As such, I increase my score to 6.

---

> > > > > ### Author Response · Authors · 2024-11-24
> > > > > **Thanks for your quick responses and raising the score!**
> > > > >
> > > > > Thank you for your quick responses and for raising the score! We are happy that our responses have addressed your concerns.
> > > > >
> > > > > Thank you again for your effort in reviewing our paper!

---

### Official Review · Reviewer_7fYQ · 2024-11-05

**Soundness:** 3
**Presentation:** 3
**Contribution:** 2
**Rating:** 5
**Confidence:** 3

**Summary:**

This paper studies both defenses and attacks to retrieval-augmented generation, which has been used in many applications. The proposed attack and defense are based on the observation that poisoning attacks tend to occur along directions for which clean data distribution has small variances.

**Strengths:**

1. The attacks and defenses to RAG  are an active research topic, given RAG is used in many real-world applications. Additionally, existing attacks are summarized in the paper.

2. Multiple attacks on RAG are considered.

3. The analysis made in the paper is interesting. For instance, Figure 1 shows some empirical evidence to verify the developed theory.

**Weaknesses:**

1. One limitation of the method is that the assumption can be strong. For instance, it is assumed that adversarial query has a different distribution from normal query. However, in practice, an attacker may select normal queries as target queries. In this scenario, the distribution of the adversarial query would be the same as the target query. This assumption may hold for certain attacks. The authors may consider narrowing down the scope, i.e., focusing on the scenarios where the adversarial query has a different distribution from the target query.

2. The assumption 1 is not very clear. How to measure the distance between two texts? The authors may consider adding more explanations to make it easier for readers to understand. Also, assumption 1 states the distance between two texts is bounded, which may not be informative, as it may hold for two arbitrary texts in practice.

3. The proposed defense may influence the utility of RAG. For instance, if new knowledge is added for a query, it can be rejected if it is substantially different from clean texts in the clean data corpus. In the experiments, it is shown that the false positive rate is very high. Is it because the clean documents are irrelevant to the protected queries? It can be helpful to perform a comprehensive analysis of the proposed defense on the influence of the utility of RAG systems. One naive defense is to reject all documents whose similarities (e.g., embedding vector similarity) are high with protected queries. The authors may consider comparing with some baselines to demonstrate the effectiveness of the proposed defenses. Additionally, the evaluation in Section 5.2 for the proposed attack is very limited.

**Questions:**

See above.

---

> ### Author Response · Authors · 2024-11-21
> **Rebuttal**
>
> We deeply appreciate the reviewers for dedicating their time and effort to reviewing our manuscript and providing insightful feedback. We are pleased that all reviewers acknowledged the novelty of our work. Furthermore, we are grateful that they considered our writing clear and our approach effective across different setups. We will integrate their suggestions into the revised version of the paper.
>
> > Q: One limitation of the method is that the assumption can be strong. For instance, it is assumed that adversarial query has a different distribution from normal ...
>
> **Response**: Thank you for your valuable suggestion regarding Assumption 1 and the insightful example you provided. We would like to clarify that our threat model, as specified in Section 2.1 of the main text, precisely addresses the scenario you mentioned and is designed to avoid related issues. Therefore, we believe Assumption 1 is both valid and reasonable within the context of our threat model. We elaborate on this point in detail in the followings.
>
> In our threat model, as specified in Section 2.1 of the main text, we consider **targeted attacks** where the attacker aims to (1) manipulate the RAG system to generate a prescribed adversarial output in response to **attacker-specified adversary queries**, and (2) ensure that the RAG responds normally to **normal queries**. In this context, the distribution of the adversary-specified queries is inherently different—and should, in fact, be significantly different—from that of normal queries. This difference is the very foundation upon which the attacker exploits the system to carry out successful attacks. If the distribution of adversarial queries were identical to that of normal queries, there would essentially be no opportunity for the attacker to manipulate the RAG to produce a specific adversarial output.
>
> Given this threat model, we believe Assumption 1 is both reasonable and valid. Within our model, there is a substantial gap between the distributions of adversarial and normal queries. As a result, it should not be difficult to inject adversarial documents into the database that are close (in terms of distribution) to the adversary queries, but far from the normal queries, so that they can be retrieved effectively when given adversary queries. And this is precisely what the Assumption 1 is describing. We will further clarify this point in the revised manuscript.
>
>
> >Q: The assumption 1 is not very clear. How to measure the distance between two texts? The authors may consider adding more explanations to make it easier for readers to understand. Also, assumption 1 states the distance between two texts is bounded, which may not be informative, as it may hold for two arbitrary texts in practice.
>
> **Response**: Thank you for your valuable suggestion regarding Assumption 1. We use the $\ell_2$ distance between the embeddings of two texts to measure the distance between them by default, as described in the Notation section of the main text. Intuitively, Assumption 1 suggests that, given an adversarial query, there (almost surely) exists a set of adversarial documents that are closer to the nearest clean documents. The current statement is primarily used to simplify the proof of the main theorem. We will revise this to make it more explicit and intuitive, while emphasizing that it does not impact the theoretical proof.
>
>
>
> >Q: The proposed defense may influence the utility of RAG. For instance, if new knowledge is added for a query, it can be rejected if it is substantially different from clean texts in the clean data corpus...
>
> **Response**: Thank you for your insightful question regarding the impact of the proposed defense on RAG utility. We agree that a trade-off between defense effectiveness and normal RAG utility is inevitable, as this trade-off is **inherent in all** detection-based problems. We note that the decision threshold in our method (in the main text) is set to result in a 1% false positive rate for clean documents. In fact, the false positive rate can be adjusted by changing the threshold as specified in Algorithm 2 in the main text. Below, we provide an ablation study by adjusting different thresholds and report their false positive rates and detection rates for the autonomous driving dataset. We can observe that with a small FPR of 0.5%, the detection rate is still very high at 0.95. This indicates that the proposed defense is effective in detecting poisoned data while maintaining a low false positive rate.
>
> Table: False positive rates and detection rates for different thresholds on the autonomous driving task.
> | FPR | .5% | 1% | 2% | 5% |
> |---|---|---|---|---|
> | Detection Rate |0.95 | 0.98 | 0.99| 0.99|

---

> > ### Author Response · Authors · 2024-11-21
> > **Rebuttal continues**
> >
> > >Q: The evaluation in Section 5.2 for the proposed attack is very limited.
> >
> > **Response**: Thank you for your valuable suggestion regarding the evaluation of our newly proposed attack. To the best of our knowledge, we are not aware of any other existing attacks (with open-sourced code) that aim to achieve similar goals, i.e., generating poisoned data with high attack success rates while explicitly forcing them to be less discernible to detection, in the context of RAG, except for the ones already included. In fact, the AgentPoison attack (first on arXiv in July this year) itself is very new, as quoted: '... the first backdoor attack targeting generic and RAG-based LLMs ...'.
> >
> > Nonetheless, we have introduced a new attack (proposed by ourselves, motivated by backdoor literature) that penalizes the Wasserstein distance between adversarial and normal queries, instead of the proposed DRS. We summarize the results in the following table, where we observe that the detection rate of the Wasserstein distance-based attack is higher than that of the proposed DRS-based attack (with a lower detection rate indicating that the attack is more effective), indicating the effectiveness of our proposed DRS attack algorithm.
> >
> > **Table**: Filtering rates for poisoned data, generated by AgentPoison and our newly proposed DRS-regularized AgentPoison, and the Wasserstein-regularized AgentPoison. The decision threshold for filtering is set to the 99th percentile of the **clean** scores, resulting in a false positive rate of approximately 1% for clean documents.
> >
> > | Attack Method                     | Perplexity filter | ℓ2-norm filter | ℓ2-distance filter | DRS (proposed) |
> > |-----------------------------------|-------------------|----------------|-------------------|----------------|
> > | AgentPoison                       | 0.03              | 0.03           | 0.01              | 0.99           |
> > | DRS-regularized AgentPoison       | 0.03              | 0.01           | 0.01              | 0.85           |
> > | Wass-regularized AgentPoison (New)| 0.03              | 0.02           | 0.01              | 0.94          |

---

### Official Review · Reviewer_pzSQ · 2024-11-18

**Soundness:** 3
**Presentation:** 2
**Contribution:** 2
**Rating:** 3
**Confidence:** 4

**Summary:**

This paper studies data poisoning attacks against Retrieval-Augmented Generation (RAG) systems. RAG systems can be compromised when attackers inject manipulated data into the retrieval corpus. The authors suggest that succesful attacks may exploit low-variance directions in the data distribution. Based on these findings, the authors introduce two significant innovations: a defense method called Directional Relative Shifts (DRS), which detects potential poisoning by analyzing shifts in low-variance directions, and a stealthier attack method that reduces detectability by minimizing DRS scores for poisoned data. The experiments show the effectiveness of the proposed defense across various RAG applications, such as Q&A systems and medical data retrieval, while the new attack algorithm succeeds in circumventing traditional and DRS defenses under specific settings.

**Strengths:**

The Directional Relative Shifts (DRS) metric is the most interesting contribution of the paper: it is a novel measure to detect poisoned documents. Moreover, Both theoretical and empirical results are provided. In terms of clarity, the paper is well written and easy to follow.

**Weaknesses:**

A significant shortcoming is the absence of reported attack success rates in the experimental results. Without this metric, it becomes difficult to fully evaluate the effectiveness of both the proposed attacks and defenses.

The paper also lacks a deep discussion on the computational cost of DRS. The access to clean documents need better justification and analysis.

**Questions:**

- How defending against poisoning in RAG settings differs from defending against, e.g., jailbreak or prompt injection attacks?

- Can the authors motivate better this assumption?: "We assume the defender has access to both the retriever and the clean data corpus. When a new test document is proposed for injection into the clean corpus, the defender calculates its DRS score (to be defined later in Eq. 3) and compares it with the scores of known clean documents." How can that clean data corpus be garanteed to not poisoned? And how many clean documents would be required so achieve such guarantee?

- Could the authors elaborate on the computational overhead of calculating DRS?

- The paper suggests that attack effectiveness is maximized by targeting low-variance directions within the data distribution. Can the authors provide more detailed empirical evidence on how such low-variance features manifest in real-world documents? Also, could you please specify the experimental settings of Fig. 2?


- A sensitivity analysis of the hyperparameters $\lambda_1$ and $\lambda_2$ would give insight into the attack’s trade-offs between attack sucess rate and evasion of the defense.

---

> ### Author Response · Authors · 2024-11-22
> **Rebuttal**
>
> We deeply appreciate the reviewers for dedicating their time and effort to reviewing our manuscript and providing insightful feedback. We are pleased that all reviewers acknowledged the novelty of our work. Furthermore, we are grateful that they considered our writing clear and our approach effective across different setups. We will integrate their suggestions into the revised version of the paper.
>
> >Q: A significant shortcoming is the absence of reported attack success rates in the experimental results. Without this metric, it becomes difficult to fully evaluate the effectiveness of both the proposed attacks and defenses.
>
> **Response**: Thank you for your insightful comment regarding the evaluation of the proposed attacks and defenses. **All the attacks are implemented by directly running the open-sourced code released by the authors without any modification.** As a result, the ASR of the attacks is the same as reported in the original papers. Specifically, all these attacks have decent ASR as reported in the original papers. For instance, AgentPoison has an ASR of around 0.8 on the autonomous driving dataset. We will include the ASR of the attacks in the revised manuscript for better clarity.
>
>
>
> >Q: The paper also lacks a deep discussion on the computational cost of DRS.
>
> **Response**: Thank you for your insightful comment regarding the computational cost of DRS. The computation and employment of DRS are very light. Specifically, we first collect a set of clean documents and obtain their embeddings. Then, we perform SVD/eigen-decomposition (**only once and can be done offline**) on the embedding matrix and store the resulting eigenvalues and eigenvectors. When new test data arrives, we only need to compute the DRS score by calculating the inner product between the test data embedding and the previously stored eigenvectors. Thus, the computation involves only a few matrix-vector multiplications and is very efficient. We will include this information in the revised manuscript for better clarity.
>
>
> >Q: The access to clean documents need better justification and analysis. How defending against poisoning in RAG settings differs from defending against, e.g., jailbreak or prompt injection attacks?
>
> **Response**: Thank you for your insightful comment regarding access to clean documents and the difference from other attacks. The main difference between poisoning attacks in RAG and the other attacks you mentioned is that RAG poisoning attacks heavily rely on the successful **retrieval of adversary-injected documents** from the database using adversarial prompts/queries. In contrast, typical adversarial attacks, like jailbreak against LLMs, do not involve retrieval and solely rely on crafting adversarial prompts/queries to lead the LLM to generate adversarial outputs.
>
> Given this discussion, access to clean documents is essential for the defender to detect adversary-injected documents. Since this is where the poisoning occurs, the defender needs access to clean documents to identify which ones are potentially poisoned. We will further clarify this point in the revised manuscript.
>
>
> >Q: Can the authors motivate better this assumption?: "We assume the defender has access to both the retriever and the clean data corpus. When a new test ...
>
> **Response**: Thank you for your insightful question regarding the clean data assumption. In fact, the defender must have access to a set of (known) clean documents in order to distinguish between future clean and potentially poisoned documents. Imagine a scenario where you are given a basket of apples, some green and some red, and you are asked to identify the red apples. Although you can divide the apples into two groups, you wouldn't know which group contains the red apples without first knowing what the red color looks like. Similarly, without knowing what a clean document looks like, it would be impossible to accurately identify the poisoned ones. This assumption is fundamental in detection-based literature, such as Out-of-Distribution detection.
>
> More fundamentally, the detection problem can be framed as a hypothesis testing or binary classification problem. In this context, the defender must understand the distribution of either clean data (which is typical in detection-based problems) or adversarial data. Without this knowledge, we can show that the optimal detector for differentiating between clean and adversarial data would effectively be reduced to randomly guessing.
>
> Regarding the number of clean documents required, if you mean "provable guarantees," such as those in conformal literature, then the number of clean documents needed to achieve such a guarantee would depend on the distribution of clean and adversarial data, which we leave as future work. In our experiments, the number of clean documents required is relatively small (e.g., 500-1000) to achieve a high detection rate. We will further clarify this point in the revised manuscript.

---

> > ### Author Response · Authors · 2024-11-22
> > **Rebuttal continues**
> >
> > >Q: The paper suggests that attack effectiveness is maximized by targeting low-variance directions within the data distribution. Can the authors provide more detailed empirical evidence on how such low-variance features manifest in real-world documents? Also, could you please specify the experimental settings of Fig. 2?
> >
> > **Response**: Thank you for your insightful question regarding the low-variance directions. We provide a real-world document example to illustrate the low-variance directions in Figure 1 in the main text. In Figure 1, we follow the exact setup in AgentPoison (Chen et al., 2024a) to generate poisoned documents by employing three different attacks: Ap, BadChain, and AutoDan, with ASR of Ap > BadChain ≥ AutoDan. Next, we plot the relative changes for different attacks. We define the relative changes in Lines 288-295. Specifically, the relative change along a certain direction of the embedding vectors is calculated by measuring the difference between adversarial and clean documents along the directions of the clean documents. This difference is then normalized by dividing the relative mean by the standard deviation along these directions. We observed that more effective attacks, such as AgentPoison, tend to have a larger relative distance along directions with small variances (i.e., the left-most group represents the directions of clean embedding documents with the top 100 smallest variances), which empirically verifies our theory.
> >
> > Regarding the experimental settings of Figure 2, it uses the same setup as Figure 1.
> >
> >
> > >Q: A sensitivity analysis of the hyperparameters and would give insight into the attack’s trade-offs between attack sucess rate and evasion of the defense.
> >
> > **Response**: Thank you for your insightful comment regarding the sensitivity analysis of the hyperparameters. We provided an ablation study on the hyperparameters of the proposed attack in the appendix and have included the table here for your reference. By using an appropriate value of $\lambda_2$, specifically 1, we can achieve a good trade-off between the attack success rate and the evasion of the defense.
> >
> > Table: Sensitivity analysis of the hyperparameters ($\lambda_2$) of the proposed attack on the autonomous driving task.
> > | λ₂   | Attack Success Rate|   DRS (proposed) Detection Rate|
> > |------|-------------------|----------------|
> > | 0.1  | 0.78           |              0.99           |
> > | 0.5  | 0.76             |            0.85           |
> > | 1    | 0.70             |       0.72           |
> > | 5    | 0.51              |            0.51           |
> > | 10   | 0.42              |             0.29           |

---

### Meta-Review · Area_Chair_KySE · 2024-12-19

**Metareview:**

This paper received three negative review and one positive review. Three reviewers pointed out one major weakness is about the strong assumption that malicious and normal queries are different. This makes the method cannot defend some easily crafted attack samples. There are other issues like presentation issues, costs, influence on normal utility, missing refs, etc. After an active rebuttal, some reviewers raised scores but no one champion this paper. The AC thinks the current version is still not ready for publication.

**Additional Comments On Reviewer Discussion:**

I think this paper has a major concern of the strong assumption raised by three reviewers. This is the main reason that I think this paper is not ready to publish. Although reviewers use straightforward examples to illustrate and authors replied with some discussion, personally, I think maybe authors can add the attacker’s knowledge in the threat model to justify about the assumption. After all, no method can defend all attacks. Maybe there are some specific scenarios that the defenders somehow have the prior knowledge that can fulfill this assumption.

---

### Decision · Program_Chairs · 2025-01-22

Reject